# Concentrations, Number of Doses, and Formulations of Aluminium Adjuvants in Vaccines: A Systematic Review with Meta-Analysis and Trial Sequential Analysis of Randomized Clinical Trials

**DOI:** 10.3390/vaccines11121763

**Published:** 2023-11-27

**Authors:** Marija Barbateskovic, Sarah Louise Klingenberg, Sara Russo Krauss, Dezhao Kong, Zhangtong Wu, Sesilje B. Petersen, Mette Kenfelt, Christian Gluud

**Affiliations:** 1Copenhagen Trial Unit, Centre for Clinical Intervention Research, The Capital Region, Copenhagen University Hospital—Rigshospitalet, 2100 Copenhagen, Denmarksarah.klingenberg@ctu.dk (S.L.K.);; 2The Evidence-Based Medicine Research Center of Traditional Chinese Medicine, Liaoning University of Traditional Chinese Medicine, Shenyang 110032, China; 3Department of Evidence-Based Chinese Medicine Research Centre, The Affiliated Hospital of Liaoning University of Traditional Chinese Medicine, Shenyang 110032, China; 4Department of Occupational and Environmental Medicine, Copenhagen University Hospital—Bispebjerg and Frederiksberg, 2400 Copenhagen, Denmark; 5Stationsvej 2, 3520 Farum, Denmark; 6Department of Regional Health Research, The Faculty of Health Sciences, University of Southern Denmark, 5230 Odense, Denmark

**Keywords:** systematic review, meta-analysis, trial sequential analysis, evidence, aluminium adjuvants, vaccines, aluminium phosphate, aluminium hydroxide, randomized clinical trials

## Abstract

Aluminium adjuvants are commonly used in vaccines to boost the effects of vaccination. Here, we assessed the benefits and harms of different aluminium adjuvants vs. other aluminium adjuvants or vs. the same aluminium adjuvant at other concentrations, administered a different number of doses, or at different particle sizes used in vaccines or vaccine excipients. We conducted a systematic review with meta-analysis and Trial Sequential Analysis to assess the certainty of evidence with Grading of Recommendations Assessment, Development and Evaluation (GRADE). We obtained data from major medical databases until 20 January 2023 and included 10 randomized clinical trials of healthy volunteers. The comparisons assessed higher vs. lower aluminium adjuvant concentrations; higher vs. lower number of doses of aluminium adjuvant; and aluminium phosphate adjuvant vs. aluminium hydroxide adjuvant. For all three comparisons, meta-analyses showed no evidence of a difference on all-cause mortality, serious adverse events, and adverse events considered non-serious. The certainty of evidence was low to very low. None of the included trials reported on quality of life or proportion of participants who developed the disease being vaccinated against. The benefits and harms of different types of aluminium adjuvants, different aluminium concentrations, different number of doses, or different particle sizes, therefore, remain uncertain.

## 1. Introduction

The benefit of a vaccine is determined by the level of protection it provides against a specific pathogen. Two means of measuring protection exist: efficacy and effectiveness.

Efficacy studies assess whether a vaccine works in controlled conditions, assessing factors like immunogenicity or its ability to stimulate an immune response in clinical trials, whilst effectiveness studies evaluate if vaccination benefits people by reducing the number of individuals with the disease in clinical trials with long-term follow-up [1].

Apart from the intended advantages, a vaccine might come with one or more adverse outcomes. These negative effects could be categorized as either non-serious, like a brief and mild headache, or severe, leading to hospitalization or even fatality. Undesirable effects might manifest shortly after administering the vaccine, such as pain at the injection site, or become apparent over time, such as autoimmune reactions. The efficacy and drawbacks of a vaccine could hinge on numerous factors, encompassing the vaccine’s constituents (such as the antigen, excipient, or adjuvants), interactions among the diverse vaccine elements, the vaccine’s manufacturing process, its overall composition, and the method of administration, dosage, and the frequency of revaccinations [2].

Adjuvants are added to vaccines to enhance an immune response and improve the effectiveness of the vaccine [3,4]. Adjuvants can offer additional advantages, including diminishing the need for frequent vaccinations, reducing the antigen dosage per vaccine, and in some cases, imparting ‘cross-clade immunity’. This refers to immunity against various clades of viruses or bacteria with distinct ancestral origins. Additionally, certain adjuvants may enhance the stability of the vaccine formulation [5,6]. Commonly employed adjuvants include various forms of aluminum salts such as aluminum phosphate, aluminum hydroxide, aluminum potassium sulfate, or amorphous aluminum hydroxyphosphate sulfate [5,7]. Aluminium adjuvants have served as adjuvants in vaccines including those combating diphtheria, tetanus, and pertussis; haemophilus influenza type B; pneumococcus conjugates; hepatitis A; and hepatitis B [8]. More recently, aluminium adjuvants have been incorporated into vaccines targeting human papillomavirus (HPV), notably within Adjuvant System 04 (aluminium hydroxide and monophosphoryl lipid A), aluminium hydroxide, or amorphous aluminium hydroxyphosphate sulfate. Aluminium adjuvants are also used in COVID-19 vaccines such as CoronaVac^®^ [9] and Sinopharm Beijing Institute of Biological Products’ COVID-19 vaccine [10] in the form of aluminium hydroxide.

A recent systematic review [11] assessed the animal literature on the clearance, biodistribution, and neuromodulatory effects of aluminium adjuvants. The review [11] revealed the lack of firm knowledge on these questions in animal studies. We recently concluded in a systematic review of 118 randomized clinical trials in humans comparing aluminium adjuvants vs. placebo or no intervention [12] that we were unable to assess the benefits or harms of aluminium adjuvants. To our knowledge, no systematic review has assessed the benefits and harms of different aluminium adjuvants vs. other aluminium adjuvants or vs. the same aluminium adjuvant at other concentrations, administered a different number of doses, or at different particle sizes used in vaccines or vaccine excipients. We, therefore, conducted such a systematic review [13].

## 2. Materials and Methods

This systematic review was conducted according to our published protocol (PROSPERO CRD42018107502) [13] using Cochrane methodology [14] and our eight-step procedure [15] for evaluating statistical and clinical significance of the meta-analysis. For a detailed description of the methods used and statistical analysis plan, we refer to our protocol [13] and the online Appendix A.

### 2.1. Criteria for Considering Trials for This Review

We included randomized clinical trials in humans comparing aluminium adjuvants according to formulation, concentration, number of doses administered, or particle size. We included the trials irrespective of publication type, publication status, language, type of outcomes reported, and year of publication. We included all trial participants regardless of sex, age, ethnicity, diagnosis, comorbidity, or country of residence.

The experimental group was defined as a vaccine or vaccine excipient including one type of aluminium adjuvant formulation (e.g., aluminium potassium sulphate, aluminium hydroxide, aluminium phosphate, aluminium hydroxyphosphate sulfate, and others) at a higher aluminium adjuvant concentration, aluminium adjuvant being administered a higher number of doses, or with a smaller aluminium particle size than in the control group. The control group was defined as the same vaccine or vaccine excipient including a different type of aluminium adjuvant formulation, a similar aluminium adjuvant at a lower aluminium adjuvant concentration, being administered a lower number of doses, or a larger aluminium particle size than in the experimental group.

We accepted co-interventions if delivered equally to the trial comparison groups.

### 2.2. Types of Outcomes

Predefined primary outcomes were all-cause mortality, proportion of participants with disease to be prevented by the vaccine, and proportion of participants with one or more serious adverse events (including individual serious adverse events analyzed separately). Secondary outcomes were health-related quality of life and proportion of participants with one or more adverse events not considered serious (including individual adverse events analyzed separately). Exploratory outcomes were serological responses. We used the trial results reported at maximum follow-up.

### 2.3. Search Methods for Identification of Trials

We searched the Cochrane Central Register of Controlled Trials in the Cochrane Library, MEDLINE Ovid, Embase Ovid, LILACS, BIOSIS, Science Citation Expanded, Conference Proceedings Citation Index—Science, Chinese Biomedical Literature Database, China Network Knowledge Information, Chinese Science Journal Database, and Wanfang Database until 20 January 2023. Search strategies are presented in the Appendix A. 

To identify grey literature and ongoing or unpublished trials, we also searched Google Scholar, the Turning Research into Practice (TRIP) Database, ClinicalTrials.gov, and the WHO International Clinical Trial Registry Platform until January 2023. In addition, we applied EMA and FDA for clinical study reports assessing any aluminium containing vaccine.

We reviewed bibliographic references of identified randomized clinical trials and review articles.

### 2.4. Data Collection

Two review authors (SRK, SLK, MB, and CG) independently screened all titles and abstracts for inclusion of potentially eligible trials in Covidence (www.covidence.org/ accessed on 23 January 2023). Reports deemed potentially relevant were obtained in full text and assessed for inclusion. Two review authors independently extracted all data using data extraction forms designed for the purpose. We contacted trial investigators or study sponsors to obtain any type of missing data. We also contacted investigators of trial protocols for which we could find no data.

Two review authors independently assessed the risk of bias of each included trial in accordance with RoB 1 in the Cochrane Handbook for Systematic Reviews of Interventions [16].

### 2.5. Measures of Treatment Effect and Data Synthesis

We performed the analyses using RevMan Web [17] and Trial Sequential Analysis version 0.9.5.6 beta [18,19]. We assessed dichotomous outcomes by calculating the risk ratio (RR) and continuous outcomes by calculating the mean difference (MD), with 95% confidence interval (CI) and the Trial Sequential Analysis-adjusted (TSA-adjusted) CI.

#### 2.5.1. Meta-Analysis

We assessed intervention effects using both fixed-effect meta-analyses [20] and random-effects meta-analyses [21]. For assessment of significance, we chose the more conservative point estimate of the two, comprising the estimate closest to zero effect [15]. If the two estimates were comparable, we used the estimate with the widest CI. For the analyses of our three primary outcomes, we considered a *p* value of less than 0.025 to be significant, as this secures a ‘family-wise’ error rate (FWER) below 0.05 [15].

#### 2.5.2. Trial Sequential Analysis

TSA-adjusted CIs were used to assess the uncertainty (risk of random errors) sparse data, multiple outcomes, and multiple testing of accumulating data [22,23,24,25,26,27], and we calculated the diversity-adjusted required information size (DARIS). We used an alpha of 2.5% for the primary outcomes and 3.3% for the secondary outcomes, a power of 90% (beta 10%), the observed proportion of events in the control group of the meta-analysis, and a diversity as suggested by the trials in the meta-analysis. As anticipated intervention effects in the TSAs, we used a realistic a priori relative risk reduction (RRR) or relative risk increase (RRI) of 20%.

#### 2.5.3. Subgroup Analysis and Sensitivity Analyses

We planned to perform a few subgroup analyses to investigate clinical heterogeneity and sensitivity analyses to assess the potential impact of the missing data. However, due to lack of data, it was not possible to conduct any subgroup analyses or sensitivity analyses assessing the impact of missing data.

#### 2.5.4. Summary of Findings and Assessment of the Certainty of the Evidence

We used the Grading of Recommendations Assessment, Development and Evaluation (GRADE) system to assess the certainty of the body of evidence associated with each of the outcomes [28]. We constructed a summary of findings table using the GRADEpro software [29].

### 2.6. Patient and Public Involvement

During the protocol stage, both a patient and a public representative actively participated in shaping the research question and defining the study outcomes. Their involvement extended to the interpretation of findings and contributed to the writing process. Furthermore, there are intentions to share the research results with the public and the pertinent patient communities as part of the dissemination plan.

## 3. Results

### 3.1. Results of the Search

The flow of records acquired through electronic searches is depicted in the Preferred Reporting Items for Systematic Reviews and Meta-Analyses (PRISMA) flowchart, available in the Appendix A. A total of 21,447 records were identified through our database searches. From the total 21,447 records identified, we removed 7787 duplicates, and we screened 13,660 titles and abstracts. We excluded 13,533 irrelevant records based on title and abstract. A total of 127 full-text reports were obtained and evaluated for eligibility. Out of these, 116 records were excluded (with reasons: 33 adjuvants other than aluminium; 33 not a randomized clinical trial; 11 different antigen concentrations; 11 vaccines control without aluminium; 3 AS04; 8 not in humans; 5 different vaccines; 3 aluminium controls without vaccine; 1 saline placebo control) (Appendix A). We classified eight trials as awaiting classification (in all of them, there was no information on the adjuvant used in the vaccine, neither was it described if the groups used different adjuvants or the same adjuvant at different concentrations).

We contacted all corresponding authors in an effort to obtain missing information or clarification regarding unclear details. Unfortunately, no additional information was received in response. Additionally, we received no clinical study reports from EMA or FDA.

### 3.2. Characteristics of Included Trials

We identified 11 records describing 10 randomized trials [30,31,32,33,34,35,36,37,38,39] (1 of the 11 records was a sub-study [40] including a total of 1848 participants that fulfilled our inclusion criteria). All included trials randomized healthy volunteers. In regard to age, the included trials randomized adults [31,35,36,37,38], adolescents [33], children [30], toddlers [32], infants [39], and a mixed population [34].

The 10 included randomized trials were published between 1983 and 2017, had a parallel-group design, and were phase 1 and/or phase 2 trials. The trials were performed in the USA [30,38,39], in Belgium [31,33,36], in multiple countries [34,35], in the Netherlands [32], and in Austria [37]. One of the trials, NCT00562237, was prematurely terminated because the interim immunogenicity results did not justify the vaccine formulations, and no data were available.

#### 3.2.1. Interventions and Comparisons

##### Types of Aluminium Adjuvants

The included trials assessed the effect of different types of aluminium adjuvants: aluminium phosphate [30,32,36,38,39], aluminium hydroxide [31,32,35,36,37], alum [30], and amorphous aluminium hydroxyphosphate sulfate [34] (Table 1).

#### 3.2.2. Vaccines against Different Viruses, Bacteria, Toxins, or Diseases

The included trials assessed the effect of vaccines against different viruses, bacteria, toxins, or diseases: influenzae [35], diptheria tetanus pertussis (dTp) [30,33], respiratory syncytial virus [38], Neisseria meningitidis [32], genital herpes [36], Streptococcus pneumoniae [39], Staphylococcus aureus [37], and hepatitis A [31]. One trial assessed the effect of higher vs. lower aluminium adjuvant concentration without an antigen [34].

#### 3.2.3. Number of Vaccinations

The included trials assessed aluminium administered at different number of doses: four times [31,39], three times [30,32,34,36], two times [32,37,38], and one time [33,38].

#### 3.2.4. Aluminium Concentrations

The included trials used different aluminium concentrations ranging from 133 microgram/dose to 1340 microgram/dose.

#### 3.2.5. Control Groups

Five trials [31,33,34,38,39] compared different concentrations of the same aluminium adjuvant, two trials [32,38] compared aluminium administered a different number of doses, and one trial [36] compared different aluminium adjuvants at the same concentration per dose.

We did not find any trials comparing aluminium adjuvants with different particle sizes. For an overview of the comparisons for the included trials, see Table 1.

#### 3.2.6. Follow-Up

Of the seven trials reporting on serious adverse events, none provided long-term follow-up data (more than one year after end of vaccination), two trials [36,38] provided data for a medium-term follow-up (1 to 12 months after end of vaccination), four trials [31,34,37,39] provided data for a short-term follow-up (less than one month after end of vaccination), and one trial [32] did not specify the follow-up of serious adverse events.

Of the six trials reporting on adverse events considered non-serious, none provided data on a long-term follow-up (more than 1 year after end of vaccination), one trial [38] provided data on a medium-term follow-up (1 to 12 months after end of vaccination), and five trials [31,33,34,36,39] provided data on a short-term follow-up (less than one month after end of vaccination).

#### 3.2.7. Sponsorship of Trials

Four trials [30,31,32,33] did not report on funding; six trials [34,35,36,37,38,39] were funded by the industry.

#### 3.2.8. Risk of Bias

Based on the information that was collected from the published reports and information from authors, we assessed all included trials at an overall high risk of bias.

### 3.3. Effects of Interventions

#### 3.3.1. Comparison 1: Higher vs. Lower Concentration of Aluminium Adjuvants

Seven trials [31,33,34,35,37,38,39] compared a higher aluminium concentration with a lower aluminium concentration (same aluminium adjuvant and same number of doses).

All-cause mortality, proportion of participants with disease to be prevented by the vaccine, and health-related quality of life were not reported.

##### Serious Adverse Events

Three trials [34,38,39] with a total of 919 participants and follow-up of 14 days [34], 28 days [39], and 270 days [38] reported on serious adverse events. In addition, two trials [31,33] reported that serious adverse events occurred, but these were not assigned per intervention group.

A total of 12/552 (2.2%) in the higher group vs. 2/367 (0.5%) in the lower group of aluminium concentration suffered from a serious adverse event. A meta-analysis showed no evidence of a difference of higher vs. lower concentration of aluminium adjuvants (risk ratio (RR) 1.99; 95% confidence interval (CI) 0.63 to 6.29; I^2^ = 0%; 919 participants; 3 trials; Figure 1).

TSA showed that less than 5% of DARIS was accrued. The certainty of evidence was very low (Table 2).

In total, 18 different types of serious adverse events were reported (Appendix A). There was no overlap of events among the trials; thus, a meta-analysis was not possible to perform.

##### Adverse Events Not Considered Serious

Five trials [31,33,34,38,39] with a total of 1645 participants and follow-up of 3 days [31], 14 days [33,34,39], and 154 days [38] reported on adverse events.

A total of 913/1024 (89.2%) in the higher group vs. 539/621 (86.8%) in the lower group of aluminium concentration suffered from an adverse event. A meta-analysis showed no evidence of a difference of higher vs. lower concentration of aluminium adjuvants (RR 1.01; 95% CI 0.97 to 1.05; I^2^ = 21%; TSA-adjusted CI 0.92 to 1.12, Figure 2). TSA showed the Z-curve (the blue line) breaching the boundary for futility and DARIS, showing that there was enough information to reject that higher compared with lower aluminium concentrations increase the risk of one or more adverse events considered non-serious by 20% (Figure 3). The certainty of evidence was low (Table 2).

In total, 25 different types of adverse events not considered serious were reported by at least two trials. The meta-analysis of each of the adverse events showed no evidence of a difference between higher vs. lower concentrations of aluminium adjuvants (Appendix A). In addition, another 81 types of adverse events not considered serious were only reported by one trial (Appendix A).

##### Serological Response

Three trials [31,33,39] with a total of 762 participants and follow-up of 4 weeks reported serological response—all with insufficient reporting of data; thus, it was not possible to conduct a meta-analysis as a primary analysis. In a sensitivity analysis, we imputed missing data (SD was calculated from CI for two trials [33,39] and imputed for one trial [31]). A meta-analysis showed no evidence of a difference in higher vs. lower concentrations of aluminium adjuvants on serological response (Appendix A).

#### 3.3.2. Comparison 2: Higher vs. lower Number of Doses of Aluminium Adjuvants

Two trials [32,38] compared the same aluminium adjuvants at the same concentration but administered a different number of times.

The proportion of participants with disease to be prevented by the vaccine and health-related quality of life were not reported.

##### All-Cause Mortality

Two trials [32,38] with a total of 672 participants and follow-up of 7 days [32] and 154 days [38] reported on all-cause mortality. Both trials reported that none of the participants died. The certainty of evidence was very low (Table 2).

##### Serious Adverse Events

Two trials [32,38] with a total of 675 participants and follow-up of 7 days [32] and 154 days [38] reported on serious adverse events.

A total of 4/341 in the group of the higher number of doses vs. 7/334 in the group of the lower number of doses experienced a serious adverse event. A meta-analysis showed no evidence of a difference in higher vs. lower number of doses of aluminium adjuvants (RR 0.62, 95% CI 0.19 to 2.00, I^2^ = 0%). TSA showed that less than 5% of DARIS was accrued. The certainty of evidence was very low (Table 2).

In addition, 17 different types of serious adverse events were reported by August et al. (Appendix A).

##### Adverse Events Not Considered Serious

Two trials [32,38] with a total of 544 participants and follow-up of 7 days [32] and 154 days [38] reported on adverse events not considered serious. However, one trial [32] reported trial results in a way that was not suitable for meta-analysis.

August et al. reported that a total of 240/272 in the group with the higher number of doses vs. 236/272 in the group with the lower number of doses experienced an adverse event not considered serious. A meta-analysis of the comparisons in the trial by August et al. showed no evidence of a difference in higher vs. lower number of doses of aluminium adjuvants (RR 1.02, 95% CI 0.91 to 1.13, I^2^ = 65%, TSA-adjusted CI 0.88 to 1.18). TSA showed the Z-curve (the blue line) breaching the boundary for futility, showing that there was enough information to reject that higher compared with lower doses increase the risk of one or more adverse events considered non-serious by 20% (Appendix A). The certainty of evidence was low (Table 2).

In addition, 26 types of adverse events not considered serious were reported (Appendix A).

##### Serological Response

One trial [32] with a total of 117 participants and a follow-up of 42 days reported on serological response. Serological response was reported as geometric mean titer (GMT).

The trial reported data insufficiently (only GMT and no SD or CI). In a sensitivity analysis, we imputed missing data (SD). De Kleijn et al. [32] found no evidence of a difference between higher vs. lower doses of aluminium adjuvant (MD −0.81, 95% CI −1.46 to −0.16) (Appendix A).

#### 3.3.3. Comparison 3: Aluminium Phosphate Adjuvant Compared with Aluminium Hydroxide Adjuvant

Two trials [32,36] compared aluminium phosphate adjuvant with aluminium hydroxide adjuvant using the same concentration/aluminium content and same number of doses.

The proportion of participants with disease to be prevented by the vaccine and health-related quality of life were not reported.

##### All-Cause Mortality

Two trials [32,36] with a total of 188 participants and a follow-up of 7 days [32] and 180 days [36] reported on all-cause mortality. Both trials reported that none of the participants died. The certainty of evidence was very low (Table 2).

##### Serious Adverse Events

Two trials [32,36] with a total of 188 participants and a follow-up of 7 days [32] and 180 days [36] reported on serious adverse events. A total of 1 (teeth extraction)/30 in the aluminium phosphate group vs. 0/30 in the aluminium hydroxide group experienced a serious adverse event in the trial by Leroux-Roels et al. [36]. The trial by de Kleijn et al. [32] reported that none of the participants experienced any serious adverse events. TSA showed that less than 5% of DARIS was accrued. The certainty of evidence was very low (Table 2).

##### Adverse Events Not Considered Serious

Two trials [32,36] with a total of 188 participants and a follow-up of 3 days [36] and 1 week [32] reported on adverse events not considered serious. However, one trial [32] reported on trial results in a way that was not suitable for meta-analysis.

Leroux-Roels et al. [36] reported that a total of 17/30 in the aluminium phosphate group vs. 15/30 in the aluminium hydroxide group experienced an adverse event not considered serious (RR 1.13, 95% CI 0.70 to 1.82, TSA-adjusted CI 0.06 to 21.04). TSA showed that only 5.2% of DARIS (60/1154 participants) was accrued. The certainty of evidence was very low (Table 2).

In addition, three different types of adverse events not considered serious were reported (Appendix A).

##### Serological Response

Two trials [32,36] with a total of 177 participants and a follow-up of 42 days [32] and 180 days [36] reported on serological response. Serological response was reported as GMT.

Both trials reported data insufficiently; thus, it was not possible to conduct a meta-analysis as a primary analysis. In a sensitivity analysis, we imputed missing data (SD was calculated from CI for one trial [36] and imputed for the other trial [32]). A meta-analysis showed no evidence of a difference in aluminium phosphate adjuvant vs. aluminium hydroxide adjuvant on serological response (Appendix A).

## 4. Discussion

We included 10 randomized trials including a total of 1848 participants. The trials included healthy volunteers of different age groups. All trials were considered at an overall high risk of bias. Our results on higher vs. lower concentrations of aluminium adjuvants, higher vs. lower numbers of doses, and aluminium phosphate vs. aluminium hydroxide showed no evidence of a difference in all-cause mortality, serious adverse events, adverse events not considered serious, and serological response. The findings are very uncertain due to scarce data and low or very low certainty of evidence.

Our review has several strengths. We seem to be the first to have conducted a systematic review assessing the benefits and harms of aluminium adjuvants used in a vaccine or a vaccine excipient vs. the same vaccine or vaccine excipient but having different concentrations, being administered at different numbers of doses, having different aluminium adjuvants formulation, or with a different aluminium adjuvant type. The work aligns with our peer-reviewed protocol [13], and we adhered to established methods recommended by Cochrane [14]. A key strength lies in our inclusive approach, encompassing trials irrespective of the publication language and whether they reported data on the pre-defined outcomes. We actively reached out to trial authors and companies to acquire both published and unpublished supplementary data. Ensuring rigor, data extraction was conducted by independent review authors in a blinded manner to minimize the risk of inaccuracies. In assessing risk of bias, we employed the Cochrane Handbook for Systematic Reviews of Interventions RoB 1 [16,41]. To gauge the certainty of the body of evidence, we utilized the GRADE framework [28,29]. These robust methodologies contribute to the credibility and thoroughness of our review.

Our systematic review is not without limitations. The quality, quantity, and reporting of outcomes in the included trials significantly impact the reliability of our findings, interpretations, and conclusions. First, our review did not include any marketed vaccines but only phase I/II trials. This is partially due to our questions being addressed primarily in dose-finding and titration studies in early-phase trials. Our search for trial results revealed that it is most likely not standard practice to publish the results of such trials revealing publication bias. Thus, a limitation of this review is that the presented results depend on the publicly available trial data only. Hence, we may lack data on both safety and immunogenicity outcomes that have never been publicly available. Second, most of the trials reported safety outcomes one to two weeks after the last dose, so the long-term effects of the different aluminium adjuvants comparisons are unknown. Third, we selected the maximum follow-up as our primary point of interest in time. This methodology prevents us from making definitive conclusions regarding the impact of aluminum adjuvants on safety and immunogenicity after each vaccine injection, particularly in trials involving multiple vaccine administrations. Fourth, our analyses on serious adverse events were affected by poor outcome reporting; two trials were excluded from analysis, although serious adverse events had occurred, but the results were not assigned per intervention group [31,33]. Contacts to authors were not successful. Fifth, we did not specifically search for observational studies reporting on harm. We acknowledge that this approach may constitute a weakness in our review, as it leans towards emphasizing short-term benefits and harms observed in randomized clinical trials. This focus raises the potential risk of overlooking late and exceptionally rare adverse effects that might be better captured in observational studies. Sixth, none of the trials provided information on the proportion of participants who contracted the disease against which they were vaccinated. Consequently, our findings related to the impact of aluminum adjuvants on immunogenicity rely on an unvalidated surrogate outcome—the serological response to the vaccine measured through various assays. Seventh, we assessed aluminium adjuvants used in vaccines only and not aluminium adjuvants in general. Thus, we did not look into allergy immunotherapy where aluminium adjuvants are also used. Lastly, we found no evidence at all on the comparison of different particle sizes of aluminium adjuvants and on the outcome health-related quality of life.

We could find no comparable systematic reviews or meta-analyses assessing the effect of different concentrations of aluminium adjuvants, different number of doses, different aluminium adjuvant formulations, or formulations with a different aluminium particle size. In 2004, Jefferson et al. [42] found low-quality evidence suggesting that vaccines with aluminium hydroxide in young children caused significantly more erythema and induration than vaccines without aluminium. In 2018, Lin et al. [43] showed lower seroprotection with H5N1 vaccines containing aluminum adjuvants when compared to their non-adjuvanted counterparts. Additionally, H5N1 vaccines with aluminum adjuvants were associated with a higher risk of experiencing pain or tenderness at the injection site during the seven days following both the initial and second vaccinations, in contrast to the non-adjuvanted versions [43]. In 2022, we conducted a systematic review examining the advantages and drawbacks of incorporating aluminum adjuvants into vaccines compared with the inclusion of a placebo or no intervention [12], and we were only able to identify low- or very low-certainty evidence suggesting the absence of benefits and increased risks of adverse events.

Aluminium is still used as the adjuvant in many ongoing vaccination programs. It is currently used as adjuvant in the world’s most used vaccine against the COVID-19. Considering the lack of good-quality evidence of the beneficial and harmful effects of adding higher or lower concentrations, numbers of doses, different types of formulations, or with a different aluminium particle size in vaccines as presented here, the relevance of this adjuvant should be investigated in future trials. Future randomized trials involving human volunteers should adhere to ethical principles rooted in the Declaration of Helsinki. The design and conduct of these trials should also align with the guidelines set forth by the International Council for Harmonisation of Technical Requirements for Pharmaceuticals for Human Use (ICH) Good Clinical Practice, ensuring consistency with relevant regulatory requirements [44,45]. Future trials should be designed in accordance with guidelines for clinical trials (SPIRIT) [46]. Moreover, their reporting should adhere to the Consolidated Standards of Reporting Trials (CONSORT) [47]. It seems advisable first to establish if the addition of aluminium adjuvants vs. placebo or nocebo offers patient-relevant benefits that are likely to trump any harms before further randomized trials comparing different concentrations, number of doses, or forms are conducted [12]. Moreover, other forms of adjuvants need to be assessed in large randomized clinical trials with adequate methodologies.

## 5. Conclusions

We found a low to very low certainty of evidence that different concentrations of aluminium adjuvants, different doses of aluminium adjuvants, and different aluminium adjuvant formulations have any beneficial or harmful effects in humans. We found no evidence assessing the effect of different particle sizes of aluminium adjuvant. The benefits and harms of different concentrations of aluminium adjuvants, number of doses, types of aluminium adjuvants, and different particle sizes of aluminium, therefore, remain uncertain.

## Figures and Tables

**Figure 1 vaccines-11-01763-f001:**
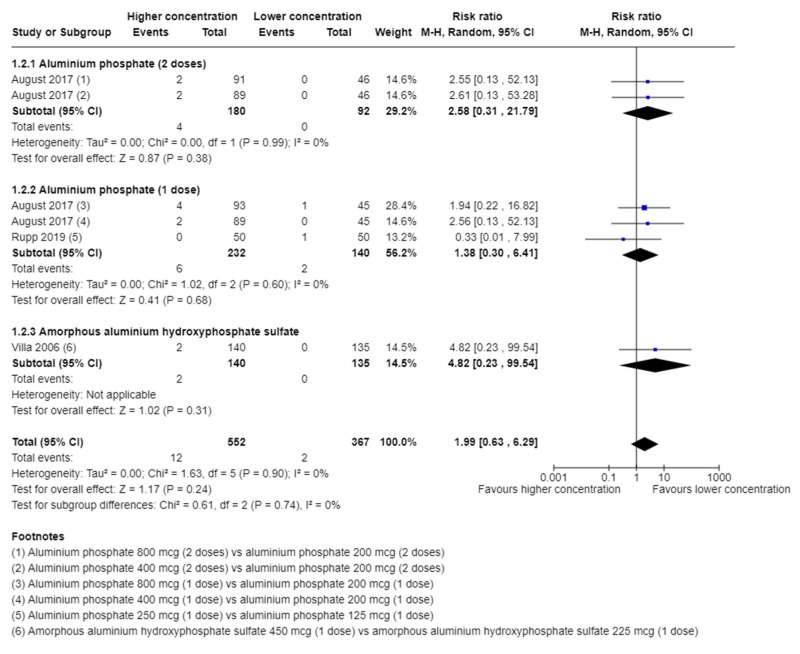
Meta-analysis of the effect of higher vs. lower concentration of aluminium adjuvant on the proportion of participants with one or more serious adverse events [34,38,39].

**Figure 2 vaccines-11-01763-f002:**
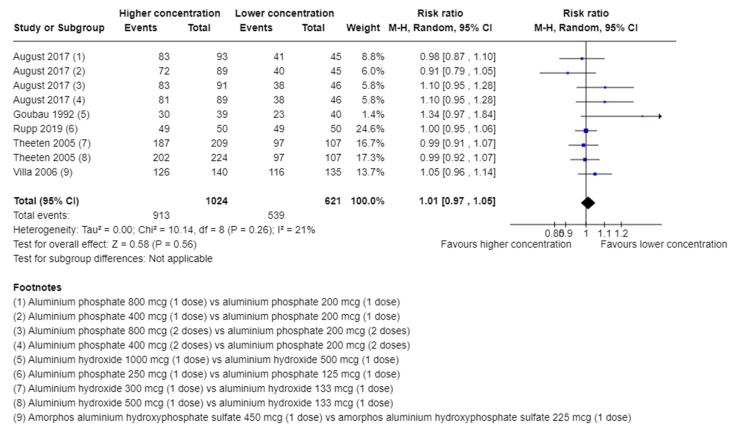
Meta-analysis of the effect of higher vs. lower concentrations of aluminium adjuvant on the proportion of participants with one or more adverse events not considered serious [31,33,34,38,39].

**Figure 3 vaccines-11-01763-f003:**
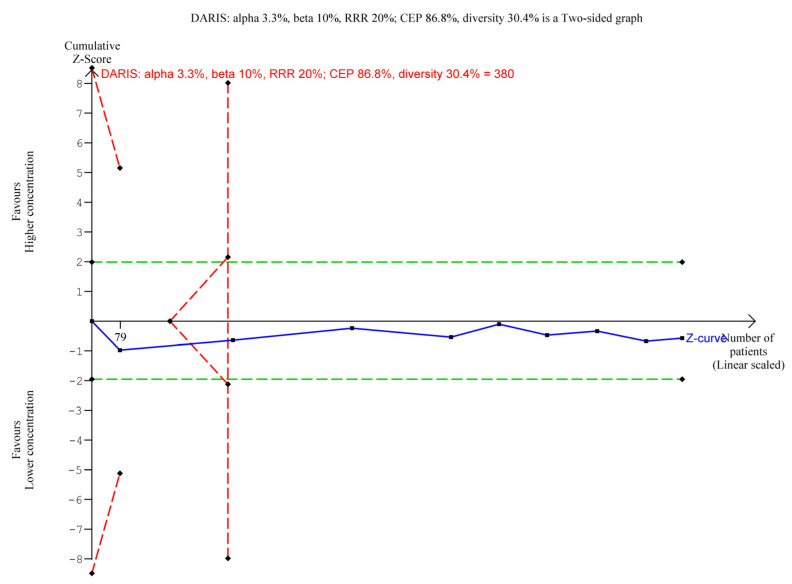
Trial Sequential Analysis of the effect of higher vs. lower concentrations of aluminium adjuvant on the proportion of participants with one or more adverse events not considered serious. We used an alpha of 3.3%, a power of 90%, control event proportion of 86.8%, diversity of 30.4%, and RRR of 20%. The relative risk was 1.01 with a TSA adjusted CI of 0.92 to 1.12.

**Table 1 vaccines-11-01763-t001:** Overview of comparisons of aluminium adjuvants in included randomized clinical trials.

Study ID	Vaccine Indication	Group 1	Group 2	Group 3	Group 4	Group 5	Group 6
August 2017 [38]	Respiratory Syncytial Virus	800 mcg AlPO_4_ (2 doses)	400 mcg AlPO_4_ (2 doses)	200 mcg AlPO_4_ (2 doses)	800 mcg AlPO_4_	400 mcg AlPO_4_	200 mcg AlPO_4_
de Kleijn, 2000 [32]	Neisseria meningitidis	860 mcg AlOH_3_ (2 doses)	860 mcg AlOH_3_ (3 doses)	1340 mcg AlPO_4_ (2 doses)	1340 mcg AlPO_4_ (3 doses)		
Goubau, 1992 [31]	Hepatitis A	1000 mcg AlOH_3_	500 mcg AlOH_3_				
Leroux-Roels, 2013 [36]	Genital herpes	500 mcg AlOH_3_	500 mcg				
Murphy, 1983 [30]	Dipteria, tetanus, pertussis (dTp)	Alum (extracted Ag)	AlPO_4_ (extracted Ag)	Alum (whole cell Ag)	AlPO_4_ (whole cell Ag)		
NCT00562237 [35]	Influenzae	1250 mcg AlOH_3_ (2 doses)	500 mcg AlOH_3_ (2 doses)				
Rupp, 2019 [39]	Streptococcus pneumonia	250 mcg AlPO_4_	125 mcg AlPO_4_				
Schwameis, 2016 [37]	Staphylococcus aureus	1000 mcg AlOH_3_	600 mcg AlOH_3_	200 mcg AlOH_3_			
Theeten, 2005 [33]	Dipteria, tetanus pertussis (dTp)	500 mcg AlOH_3_	300 mcg AlOH_3_	133 mcg AlOH_3_			
Villa, 2006 [34]	No antigen	450 mcg AAHS	225 mcg AAHS				

AAHS: aluminium hydroxyphosphate sulfate; Ag: antigen; AlOH_3_: aluminiumoxyhydroxide; AlPO_4_: aluminiumhydroxyphosphate; mcg: microgram.

**Table 2 vaccines-11-01763-t002:** Summary of findings.

Outcomes	Anticipated Absolute Effects ** (95% CI)		Relative Effect (95% CI)	No of Participants (Trials)	Certainty of the Evidence (GRADE)
	Risk with Lower Concentration or Doses, or Aluminium Hydroxide	Risk with Higher Concentration or Doses, or Aluminium Phosphate			
Comparison 1—Higher versus lower concentration of aluminium adjuvants					
All-cause mortality *	-	-	-	-	-
Participants with disease being vaccinated against *	-	-	-	-	-
Proportion of participants with serious adverse events	5 per 1000	11 per 1000 (3 to 34)	RR 1.99 (0.63 to 6.29)	919 (3 RCTs)	⨁◯◯◯ Very low ^a,b,§^
Health-related quality of life *	-	-	-	-	-
Proportion of participants with adverse events not considered serious	868 per 1000	877 per 1000 (842 to 911)	RR 1.01 (0.97 to 1.05)	1645 (5 RCTs)	⨁⨁◯◯ Low ^a,§^
Comparison 2—Higher versus lower number of doses of aluminium adjuvants					
All-cause mortality	not pooled	not pooled	not pooled ^‡^	672 (2 RCTs)	⨁◯◯◯ Very low ^a,c,§^
Participants with disease being vaccinated against *	-	-	-	-	-
Proportion of participants with serious adverse events	21 per 1000	13 per 1000 (4 to 42)	RR 0.62 (0.19 to 2.00)	675 (2 RCTs)	⨁◯◯◯ Very low ^a,b,§^
Health-related quality of life *	-	-	-	-	-
Proportion of participants with adverse events not considered serious	868 per 1000	0 per 1000 (0 to 0)	not estimable	544 (1 RCT)	⨁◯◯◯ Very low ^a,d,§^
Comparison 3—Aluminium phosphate adjuvant versus aluminium hydroxide adjuvant					
All-cause mortality	not pooled	not pooled	not pooled ^‡^	188 (2 RCTs)	⨁◯◯◯ Very low ^a,c,§^
Participants with disease being vaccinated against *	-	-	-	-	-
Proportion of participants with serious adverse events	10 per 1000	31 per 1000 (1 to 730)	RR 3.00 (0.13 to 70.83)	188 (2 RCTs)	⨁◯◯◯ Very low ^a,b,§^
Health-related quality of life *	-	-	-	-	-
Proportion of participants with adverse events not considered serious	500 per 1000	0 per 1000 (0 to 0)	not estimable	60 (1 RCT)	⨁◯◯◯ Very low ^a,d,§^

** The risk in the intervention group, along with its 95% confidence interval (CI), is derived from the presumed risk in the comparison group and the relative effect of the intervention, including its 95% CI. In this context, CI stands for confidence interval, and RR represents the risk ratio. The GRADE Working Group’s grades of evidence are as follows: High certainty: We possess a high level of confidence that the actual effect closely aligns with the estimated effect; Moderate certainty: We hold a moderate level of confidence in the effect estimate; the true effect is likely close to the estimated effect, but there exists a possibility of substantial variation; Low certainty: Our confidence in the effect estimate is limited; the true effect may significantly differ from the estimated effect; Very low certainty: We harbor very little confidence in the effect estimate; the true effect is likely to be substantially different from the estimated effect. Explanations: ^a^ Downgraded two levels for very serious risk of bias due to all trials being at an overall high risk of bias. ^b^ Downgraded three levels for extremely serious imprecision due to the accrued information size being less than 5% of the required information size. ^c^ Downgraded two levels for very serious imprecision as data provided by two small trials. ^d^ Downgraded two levels for very serious imprecision as data provided by one small trial. * No trials reported on this outcome. ^§^ As the number of trials were less than ten per outcome, we could not assess potential asymmetry in funnel plots. We could not assess the influence of vested interest in subgroup analysis. Accordingly, we could not use the trial results to assess the likelihood of publication bias and, therefore, we did not downgrade the certainty of evidence for risk of publication bias. ^‡^ Zero events in both arms.

## Data Availability

All data relevant to the study are included in the article or uploaded as Appendix A. Data sharing not applicable as all data are available in figures and tables in text or Appendix A.

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
