# Peer review of "Concentrations, Number of Doses, and Formulations of Aluminium Adjuvants in Vaccines: A Systematic Review with Meta-Analysis and Trial Sequential Analysis of Randomized Clinical Trials"

_vaccines, 2023, doi:10.3390/vaccines11121763_

Round 1

Reviewer 1 Report

Comments and Suggestions for Authors

This is a well-conducted systematic review with meta-analysis and Trial Sequential Analysis assessing the benefits and harms of aluminium adjuvants in vaccines. The focus of this review is on the concentrations, number of doses, and formulations of aluminium adjuvants in vaccines. The authors have compared higher vs lower aluminium adjuvant concentrations; higher vs lower number of doses of aluminium adjuvant; and aluminium phosphate adjuvant vs aluminium hydroxide adjuvant. Meta-analyses have shown no evidence of a difference on all-cause mortality, serious adverse events, and adverse events considered non-serious for all three comparisons. However, the certainty of evidence was low to very low. The findings of this review suggest that the concentration and formulation of aluminium adjuvants used in vaccines may not significantly impact mortality or serious adverse events. However, the results should be interpreted with caution due to the low to very low certainty of evidence.

One of the key limitations in this review is the small number of trials included. The authors have included only 10 trials in their analysis, which may not provide a robust basis for their conclusions.

Author Response

This is a well-conducted systematic review with meta-analysis and Trial Sequential Analysis assessing the benefits and harms of aluminium adjuvants in vaccines. The focus of this review is on the concentrations, number of doses, and  formulations of aluminium adjuvants in vaccines. The authors have compared higher vs lower aluminium adjuvant concentrations; higher vs lower number of doses of aluminium adjuvant; and aluminium phosphate adjuvant vs aluminium hydroxide adjuvant. Meta-analyses have shown no evidence of a difference on all-cause mortality, serious adverse events, and adverse events considered non-serious for all three comparisons. However, the certainty of evidence was low to very low. The findings of this review suggest that the concentration and formulation of aluminium adjuvants used in vaccines may not significantly impact mortality or serious adverse events. However, the results should be interpreted with caution due to the low to very low certainty of evidence.

One of the key limitations in this review is the small number of trials included. The authors have included only 10 trials in their analysis, which may not provide a robust basis for their conclusions.

Author response: We thank the reviewer for the very positive feedback. We agree that inclusion of only 10 trials is a limitation – not a limitation introduced by our methods, but a limitation caused by lack of publicly available results of trials assessing the effects of different aluminium adjuvants that may guide in choosing an aluminium adjuvant over another. We have now tried to make this even clearer in our reporting.

Reviewer 2 Report

Comments and Suggestions for Authors

My report:

Manuscript ID: vaccines-2689401

Title: Concentrations, number of doses, and formulations of aluminum adjuvants in vaccines. A systematic review with meta-analysis and Trial Sequential Analysis of randomised clinical trials

Authors: Marija Barbateskovic, Sarah Louise Klingenberg, Sara Russo Krauss, De Zhao Kong, Zhang Tong Wu, Sesilije B Petersen, Mette Kenfelt, and Christian Gluud

Overview and general recommendation:

The review proposed an assessment of the benefits and harms of different aluminium adjuvants vs. other aluminium adjuvants or vs. the same aluminium adjuvant at other concentrations, administered in a different number of doses or at different particle sizes used in vaccines or vaccine excipients. The authors conducted a systematic review with meta-analysis and trial-sequence Sequential Analysis assessing the certainty of evidence with Grading of Recommendations Assessment, Development, and Evaluation (GRADE) using the data from major medical databases until 20 January 2023. 

Overall, the study is well-designed and written. The methods are fully described, and the results are well presented and supported by statistical data analysis.

Minor thing: I think the word “analysis” is incorrectly spelled several times in the manuscript. Please revise.

Comments on the Quality of English Language

English is fine. No issues detected

Author Response

Overview and general recommendation:

The review proposed an assessment of the benefits and harms of different aluminium adjuvants vs. other aluminium adjuvants or vs. the same aluminium adjuvant at other concentrations, administered in a different number of doses or at different particle sizes used in vaccines or vaccine excipients. The authors conducted a systematic review with meta-analysis and trial-sequence Sequential Analysis assessing the certainty of evidence with Grading of Recommendations Assessment, Development, and Evaluation (GRADE) using the data from major medical databases until 20 January 2023.

Overall, the study is well-designed and written. The methods are fully described, and the results are well presented and supported by statistical data analysis.

Minor thing: I think the word “analysis” is incorrectly spelled several times in the manuscript. Please revise.

Author response: We thank the reviewer for the very positive feedback. Regarding the word “analysis” – we use analysis in singular and analyses in plural.

Reviewer 3 Report

Comments and Suggestions for Authors

The presented work is quite interesting, as it covers the results of a study of the benefits and harms of various aluminum adjuvants in comparison with other aluminum adjuvants or in comparison with the same aluminum adjuvant in other concentrations, with the introduction of different numbers of doses or with different particle sizes used in vaccines or adjuvants vaccine substances. At the same time, despite the high scientific and practical significance, the authors should answer the reviewer’s questions, after which a decision can be made on the final status of the article and consideration of the possibility of its acceptance for publication.

1. The introduction needs to be significantly improved and expanded, since as presented it is very weak and uninformative.

2. The authors of this article should provide more explanation on how exactly the selection of research objects was made in Table 1. And also why all groups were not selected for all selected research objects.

3. The results of the presented Meta-analysis should be considered in comparison with other similar works.

4. Based on the presented analysis, the authors should consider and propose other types of materials used for vaccination as alternative materials.

5. In general, the presented article requires significant revision, since in its presented form it resembles more of an analytical report on the experiments performed with a large number of subparagraphs, some of which consist of one or two paragraphs. From which it follows that the authors can combine a number of points together and reduce the number of sub-points.

Author Response

The presented work is quite interesting, as it covers the results of a study of the benefits and harms of various aluminum adjuvants in comparison with other aluminum adjuvants or in comparison with the same aluminum adjuvant in other concentrations, with the introduction of different numbers of doses or with different particle sizes used in vaccines or adjuvants vaccine substances. At the same time, despite the high scientific and practical significance, the authors should answer the reviewer’s questions, after which a decision can be made on the final status of the article and consideration of the possibility of its acceptance for publication.

Author response: We thank the reviewer for the very positive feedback

1. The introduction needs to be significantly improved and expanded, since as presented it is very weak and uninformative.

Author response: We thank the reviewer for this comment. The aim of our systematic review is to assess the evidence for the use of different aluminium adjuvants vs. other aluminium adjuvants or vs. the same aluminium adjuvant at other concentrations, administered in a different number of doses, or at different particle sizes used in vaccines or vaccine excipients. The introduction shortly introduces the reader to the field and argues why this systematic review is
needed. A more in-depth introduction can be found in our protocol published in The Cochrane Library. An exhaustive introduction to a research field is usually not the aim of a systematic review/research paper. We have now expanded the introduction and we are of course willing to expand it more if this is requested by the Editor.

“The benefit of a vaccine is determined by the level of protection it provides against a specific pathogen. Two means of measuring protection exist: efficacy and effectiveness. Efficacy studies assess whether a vaccine works in controlled conditions, assessing factors like immunogenicity or its ability to stimulate an immune response in clinical trials, whilst effectiveness studies evaluate if vaccination benefits people by reducing the number of individuals with the disease in clinical trials with long-term follow-up [1]. In addition to its intended effect, a vaccine may be accompanied by one or more harmful effects upon administration. Harms may be considered non-serious, such as mild transient headache, or serious, leading to hospitalisation or even death. Adverse effects may appear shortly after vaccine administration, like pain at the injection site, or emerge after some time, such as autoimmune responses. Vaccine toxicity, efficacy, and effectiveness may originate from, or depend on, a plethora of factors, including the vaccine components (e.g., the antigen, excipient, or adjuvant); interaction between different vaccine components; vaccine manufacture; overall vaccine composition; route of administration; dosage; and the number of booster vaccinations [2].”

2. The authors of this article should provide more explanation on how exactly the selection of research objects was made in Table 1. And also why all groups were not selected for all selected research objects.

Author response: We thank the reviewer for this comment. The search for trials and the following literature screening and selection of trials is described in details in the methods and results section. Search strategies and flowchart are included in the supplement material. The reason why not all selected potential research objects were covered is due to missing published relevant randomised clinical trials. This has now been stated explicitly in our discussion.

3. The results of the presented Meta-analysis should be considered in comparison with other similar works.

Author response: We thank the reviewer for this comment. The second last paragraph in the discussion, line 437-450, discusses our results in comparison to other similar works.

“We could find no comparable systematic reviews or meta-analyses assessing the effect of different concentrations of aluminium adjuvants, different number of doses, different aluminium adjuvant formulations, or formulations with a different aluminium particle size. In 2004, Jefferson et al. [45] found low-quality evidence that suggested that in young children, vaccines with aluminium hydroxide caused significantly more erythema and induration than vaccines without aluminium and significantly fewer reactions of all types. In 2018, Lin et al. [46] showed inferior seroprotection with aluminium-adjuvated H5N1 vaccines compared with that conferred by non-adjuvanted counterparts. Furthermore, H5N1 vaccines with aluminium adjuvants were associated with a significantly higher risk of pain/tenderness at the injection site during the seven days after the first vaccination and after the second injection vs the non-adjuvanted counterparts [46]. In 2022, we reviewed the benefits and harms of adding aluminium adjuvants to vaccines vs placebo or no intervention [10], and we were only able to identify low or very low certainty evidence suggesting absence of benefits and increased risks of adverse events.”

4. Based on the presented analysis, the authors should consider and propose other types of
materials used for vaccination as alternative materials.

Author response: We thank the reviewer for this comment. The aim of your systematic review is to assess the evidence on different concentrations of aluminium adjuvants, different number of doses, different aluminium adjuvant formulations, or formulations with a different aluminium particle size in vaccines and communicate these findings. Since we have not assessed the evidence of aluminium adjuvants compared with other types of adjuvants, we are not able to propose an alternative. We have now added a sentence about this important point.

“Aluminium is still used as the adjuvant in many ongoing vaccination programmes. It is currently used as adjuvant in the world's most used vaccine against the COVID-19. Considering the lack of good quality evidence of the beneficial and harmful effects of adding higher or lower concentration, number of doses, different types of formulations, or with a different aluminium particle size in vaccines as presented here, relevance of this adjuvant should be investigated in future trials. Future randomised trials in humans should be conducted according to the ethical principles that have their origin in the Declaration of Helsinki, and that are consistent with the ICH Good Clinical Practice and the applicable regulatory requirement(s) [47, 48]. Such trials should be designed in accordance with guidelines for clinical trials (SPIRIT) [49] and reported in accordance with the Consolidated Standards of Reporting Trials (CONSORT) [50]. It seems advisable first to establish that addition of aluminium adjuvants vs placebo or nocebo offers patient-relevant benefits that are likely to trump any harms before further randomised trials comparing different concentrations, number of doses, or forms are subjected to further randomised trials [10]. Moreover, other forms of adjuvants need to be assessed in large randomised clinical trials with adequate methodologies.”

5. In general, the presented article requires significant revision, since in its presented form it resembles more of an analytical report on the experiments performed with a large number of subparagraphs, some of which consist of one or two paragraphs. From which it follows that the authors can combine a number of points together and reduce the number of sub-points.

Author response: We thank the reviewer for this comment. A systematic review is in fact an
analytical report. We present data on two research questions which necessitate a stringent and
consequent phrasing.

Round 2

Reviewer 3 Report

Comments and Suggestions for Authors

The authors answered all the questions posed, the article can be accepted for publication.